# ANALYSING THE EFFECTIVENESS OF DIFFERENT OFFSHORE MAINTENANCE BASE OPTIONS FOR FLOATING WIND FARMS

Nadezda Avanessova[a], Anthony Gray[b], Iraklis Lazakis[c], R Camilla Thomson[d], and Giovanni Rinaldi[e]

[a]Industrial CDT in Offshore Renewable Energy (IDCORE), University of Strathclyde, Glasgow, G1 1XQ, UK
[b]Offshore Renewable Energy Catapult, Glasgow, G1 1RD, UK
[c]Department of Naval Architecture, Ocean and Marine Engineering, University of Strathclyde, Glasgow, G1 1XQ, UK
[d]School of Engineering, Institute for Energy Systems, The University of Edinburgh, Faraday Building, The King's Buildings, Edinburgh, EH9 3DW, UK
[e]Renewable Energy group at the University of Exeter, Penryn Campus, Treliever Road, TR109FE, Penryn, Cornwall, UK

**Correspondence:** Nadezda Avanessova (nadezda.avanessova@ore.catapult.org.uk)

**Abstract.** With the growth of the floating wind industry, new Operation & Maintenance (O&M) research has emerged evaluating tow-to-port strategies (Offshore Wind Innovation Hub, 2020) but limited work has been done on analysing other logistical strategies for offshore floating wind farms. In particular, what logistical solutions are the best for farms located far offshore, that cannot be reached by crew transfer vessels (CTVs)? Previous studies have looked at the use of Surface Effect Ships (SES) and CTVs during the Operation and Maintenance (O&M) of bottom-fixed wind farms but only some of them included Service Operation Vessels (SOV). This study analyses two strategies that could be used for floating wind farms located far from shore using ORE Catapult's in-house O&M simulation tool. One strategy comprises of having a SOV performing most of the maintenance on the wind farm and the other strategy uses an Offshore Maintenance Base (OMB) instead which would be located next to the offshore substation and would accommodate three CTVs. This paper provides an overview of the tool and the inputs used to run it, including failure rates of floating wind turbine subsea components and their replacement costs. In total six types of simulations were run with two strategies, two different weather limits for CTVs and two weather datasets ERA5 and ERA-20C. The results of this study show that the Operational Expenditure (OPEX) costs for the strategy with an OMB are 5-8 % (depending on the inputs) lower that with SOV but if Capital Expenditure (CAPEX) costs are included in the analysis and the Net Present Value (NPV) is taken into account then the fixed costs associated with building the offshore maintenance base have a significant impact on selecting a preferred strategy. It was found that for the case study presented in this paper the OMB would have to share the foundation with a substation in order to be cost competitive with the SOV strategy.

## 1 Introduction

Floating wind is an industry with huge growth potential. There is currently 150 MW installed worldwide, with 70 GW expected to be installed by 2040 globally (Spearman et al., 2020). Floating substructures allow turbines to be installed in deeper seas and further from shore than bottom-fixed sites, which raises accessibility issues during Operation and Maintenance (O&M) of

such turbines. Currently, Operational Expenditure (OPEX) of bottom-fixed offshore wind farms contribute to up to a third of the Levelized Cost of Energy (Feng et al., 2010; Musial and Ram, 2010; Maples et al., 2013) and that proportion is expected to stay significant for floating wind farms, particularly as projects increase in size, capacity and distance to shore. Not only is the transition time increased, making Crew Transfer Vessels (CTVs) sent from shore for maintenance unfeasible because of a large proportion of workers' shifts spent on board, but also the amount of suitable weather windows is significantly reduced. Floating wind turbine foundations require at least 50 m in water depth, while their bottom-fixed alternatives can be installed in waters only below 70 m in depth. ScotWind, which is a seabed leasing round run by Crown Estate Scotland, has identified several locations for the future deployment of offshore wind. According to the analysis performed by Gray (2021a), the majority of these locations will have water depths of 70 m and more meaning that it would only be possible to install wind turbines on floating foundations. One of these areas, NE8, was selected for this study to model a realistic scenario because 100% of this area has over 70 m water depth. This area was of particular interest for O&M analysis presented in this paper because it is located 100 km from shore, meaning that maintenance vessels deployed there would spend a lot of time in transit. It is recommended that for wind farms more than 70 km from the O&M port, offshore accommodation solutions need to be used (Hu and Yung, 2020). In this study only semi-submersible floating foundations are considered due to the lack of data for other types of floaters.

This study provides a review of two logistical strategies which may be beneficial for floating wind farms. In particular, this work looks at offshore floating and fixed bases to reduce the need for long technician transfers between an O&M base port and a farm. Floating bases in this study refer to advanced Service Operation Vessels (SOVs) with all-in-one facilities; accommodation, walk-to-work gangway, maintenance and spare parts platform and a launch and recovery system for a daughter craft. A fixed base considered in this study is similar in its facilities, it accommodates three CTVs and personnel. The fixed base is further referred in text as an Offshore Maintenance Base (OMB). OMB can either share the foundation with the substation or have a separate foundation and connect to the substation via a bridge. It is likely that the emergency recovery system and the helicopter base would be shared between the substation and the accommodation base. This work investigates whether these two logistical solutions can positively affect Key Performance Indicators (KPIs) of O&M, such as cost, availability and carbon emissions, using ORE Catapult's internal O&M simulation tool COMPASS (Combined Operations and Maintenance, People, Assets and Systems Simulation).

This work exploits the time-domain mode capabilities of COMPASS. At each time step, the software calculates the revenue produced by each turbine and activates any maintenance processes depending on vessel and personnel availability. Failure rates, which are currently obtained from a combination of public reports (e.g. (Carroll et al., 2016; SPARTA, 2017; Warnock et al., 2019)) and internal expertise, were modified to represent floating wind device failures. Costs, availability and carbon emissions will be measured for two logistical setups and the results will support the decision making process during planning of substations for floating farms and O&M fleet.

The Literature Review section presents the current literature and critical review on O&M tools. The Methodology section describes some of the methods used to simulate the discussed scenarios. The Inputs and Assumptions section provides the

inputs used in this study e.g. weather data. The Results section provides the comparison of the simulated scenarios and the last section concludes this study.

## 2 Literature Review

O&M simulation tools can be used to model the O&M phase of the wind farm lifetime. There are three types of decisions that these tools can support the user with (Michael Welte et al., 2018):

- Operational: decisions that are made on a daily basis, e.g. in what order to repair failed turbines and what route to select (Lazakis and Khan, 2021)

- Tactical: decisions that are made on a yearly basis, e.g. when is the best time to deploy a jack-up vessel

- Strategic: decisions that are made before the farm starts its operation or every 10-20 years e.g. to estimate OPEX costs for the entire lifetime of a farm. Deciding between having an OMB or the SOV or a combination of the two is a strategic type of decision.

O&M simulation tools for long-term logistical planning i.e. strategic type of O&M tools and OPEX estimation typically adopt a time-domain approach. That means that at each timestep the software tool models failures and replicates the real-world decisions made in terms of undertaking unplanned maintenance and models planned maintenance according to the specified frequency. This process runs until the final timestep is reached, which replicates the end of the wind turbine life cycle (Gray, 2020). Rinaldi et al. (2021) demonstrated how detailed operation and maintenance models could be used for better estimation of operational costs of floating wind farms. Apart from OPEX other KPIs can be measured with the use of such tools e.g. energy availability and carbon emissions based on vessel usage. Accurate results require accurate inputs, where most important are failure rates and repair durations. Due to the sensitivity of this data most failure databases are anonymous and in some cases normalised (Michael et al., 2011). Most publicly available sources summarised and compared in Cevasco et al. (2021) contain failure data only for bottom-fixed wind turbines up to 2 MW in capacity and due to the differences in data collection, the results are often conflicting. Currently there is no database containing failure rates for floating wind turbines and their substructures, therefore this study utilised failure rates from oil & gas and ship industries discussed in Sect. 4.3.

The concept of an O&M platform which includes an accommodation module, a full service base and a fully operational offshore heliport has been presented in the news in 2015 by Fred. Olsen Windcarrier (Fred. Olsen, 2015). They performed O&M simulations in collaboration with a major developer in the UK, however their studies are not publicly available and the assumptions or the software used in these studies are unknown. They have claimed however to achieve the best results (98% availability) with the scenario employing three CTVs combined with an offshore accommodation platform (Fred. Olsen, 2016). Horns Rev 2 wind farm in Denmark located 32 km from the shore has also used this concept with a limit of 24 technicians on the platform however the number of CTVs used is unknown. Two wind farms in Germany, Global Tech 1 and Dan Tysk, also have offshore accommodation for 34 and 50 technicians respectively but the number of CTVs used is unknown (Echavarria et al., 2015). According to the 2018 development plans, Hornsea Three is intending to install up to three offshore accommodations

close to the farm however the characteristics of the accommodations are not known (DONG Energy, 2017). At the same time the SOV technology is in high demand for the future multi-gigawatt projects. For example, North Star has recently received an order for three SOVs to be used in the Dogger Bank Farm in the South-East UK (reNEWS.biz, 2021) which was later updated to four SOVs (Durakovic, 2021). One of them, with the capacity to accommodate 78 crew members, will perform scheduled maintenance and the other three will accommodate 60 persons each and will be used for corrective maintenance activities. There is also a push towards decarbonisation of vessels, some new generation SOVs promise to be hydrogen and electrically powered e.g. Edda Wind (Buljan, 2021) has recently launched its first commissioning SOVs which has been designed to accommodate hydrogen technology in the future. Another example is a SOV, recently developed by Siemens Gamesa, equipped with batteries and propulsion technology that can run on hydrogen (Russel, 2021).

StrathOW-OM tool (Strathclyde University strategic O&M tool) has been used to analyse the strategies with motherships, floatels and fixed accommodation (Dalgic et al., 2015). The results indicated that fixed accommodation platforms and motherships can be beneficial for the O&M of wind farms in far offshore locations. Innovative mothership concept which can deploy CTVs has also been proposed for wind farms located at a significant distance to the O&M port (Mccartan et al., 2015). There has been only a handful of studies analysing the usage of SOVs for offshore wind farms (Endrerud et al., 2015). Other studies simulated SOVs using operational O&M tools. The study performed by Besnard et al. (2013) has also compared OPEX and availability between onshore and offshore maintenance base scenarios, however this study only looked at the use of CTVs. It found that unless the work shifts are extended, having an offshore maintenance base is not cost effective because the farm availability increases together with the OPEX. There has been some work done analysing the tow-to-port strategy (Offshore Wind Innovation Hub, 2020; Rinaldi et al., 2020) for floating wind farms with conflicting conclusions on whether tow-to-port is a beneficial strategy or not, which is possibly due to the different number of wind turbines and locations considered in these studies. This study will assume all maintenance is performed offshore, however tow-to-port should be investigated further in the future work to investigate the effectiveness of this strategy. Unlike other O&M tools COMPASS can also simulate SOVs with daughter crafts and take into account the accommodation capacity of SOVs and general personnel capacities of other vessels. It can also model the return of personnel to SOVs and back to ports and calculate associated costs and carbon emissions. Storage capabilities for spare parts of SOV and OMB are not taken into account in this study; however for SOV storage optimisation; readers may refer to the study by Neves-Moreira et al. (2021) made with a tactical and operational type of O&M tool. A sensitivity analysis of offshore wind farm O&M cost and availability performed by Martin et al. (2016) has shown that one of the seven important factors affecting availability is the maintenance resources availability, however in the study maintenance resource is related to vessel mobilisation. Spare part availability or vessel storage capacity have not been analysed as separate factors there, yet they would cause a delay in O&M and are likely to affect the farm availability on a similar scale to vessel mobilisation.

## 3 Methodology

### 3.1 COMPASS tool overview

ORE Catapult's internal O&M simulation model COMPASS, has been used for this study. The COMPASS tool is a Python-based model which is interfaced with Microsoft Excel for the input and output files. The COMPASS tool is intended primarily for obtaining reliable estimates of OPEX of offshore renewable energy farms for informing internal cost modelling projects. It takes key characteristics of an offshore wind farm, such as number of turbines, site capacity etc., and applies a series of O&M activities. Figure 1 shows a high level structure of the COMPASS tool operating in a time-domain mode.

The 'bottom-up' O&M reference activities are considered with respect to the Reference Designation System for Power Plants (RDS-PP) methodology of component assignment down to the subsystem designation. O&M activities are populated within these subsystems for wind turbine components. Activities for the windfarm assets are assigned at a system level (e.g. export cables) or where a clear designation could not be determined (e.g. seabed survey) at a windfarm level.

   Each maintenance activity can either be planned (i.e. recurring scheduled measures which are organised in advance, along

with the resources required to carry out an O&M task) or unplanned (i.e. the result of a turbine alarm or component failure) and has been assigned a rate (i.e. times per year per turbine for scheduled maintenance, or failures per year per turbine for corrective maintenance). These rates have been defined from a combination of publicly available academic sources (particularly (Carroll et al., 2016) and (SPARTA, 2017)) as well as from the experience of the team at ORE Catapult. Each turbine in COMPASS is split into subsystems e.g. Generator System or Control and Protection System. Each subsystem is then split into components.

There is a set of planned and unplanned activities applied on turbine, subsystem and component level. Type of activities, their rates (with exception of activities for which failure data was publicly available) their duration, personnel and vessel requirement and urgency have been defined by the ORE Catapult team and are not available to the readers. The only exception is activities for which characteristics such as failure rate (for unplanned) or frequency (for planned) were publicly available. For the purposes of COMPASS, activities informed by the evaluation of condition monitoring systems or other relevant component

data were not separately distinguished.

   At each time step in the time-domain mode, the COMPASS tool checks whether maintenance is required and then checks for availability of personnel and vessels and suitable weather conditions (wave height and wind speed). In the case of unplanned maintenance, the Monte Carlo method is applied in order to model failures. The essence of this method is to generate a random number between 0 and 1 at each timestep and based on this number determine whether the component has failed by comparing

it to the reliability of component (i.e. the probability of not failing, 'R'). If this random number is below R, then component has failed, if it is above, the component retains its operational state.

   At the end of the simulation, the COMPASS tool sums up all the costs from all timesteps associated with O&M activities and adds fixed costs (e.g. IT and Onshore Base costs). In addition to all costs, the tool calculates total energy produced by the project and calculates both time-based and energy-based availabilities; where time-based availability is the proportion of time

that a farm was operational and energy-based availability is the proportion of energy that was generated out of the theoretical energy production(if the entire project was operating at full capacity 100% of the time). In addition to these conventional

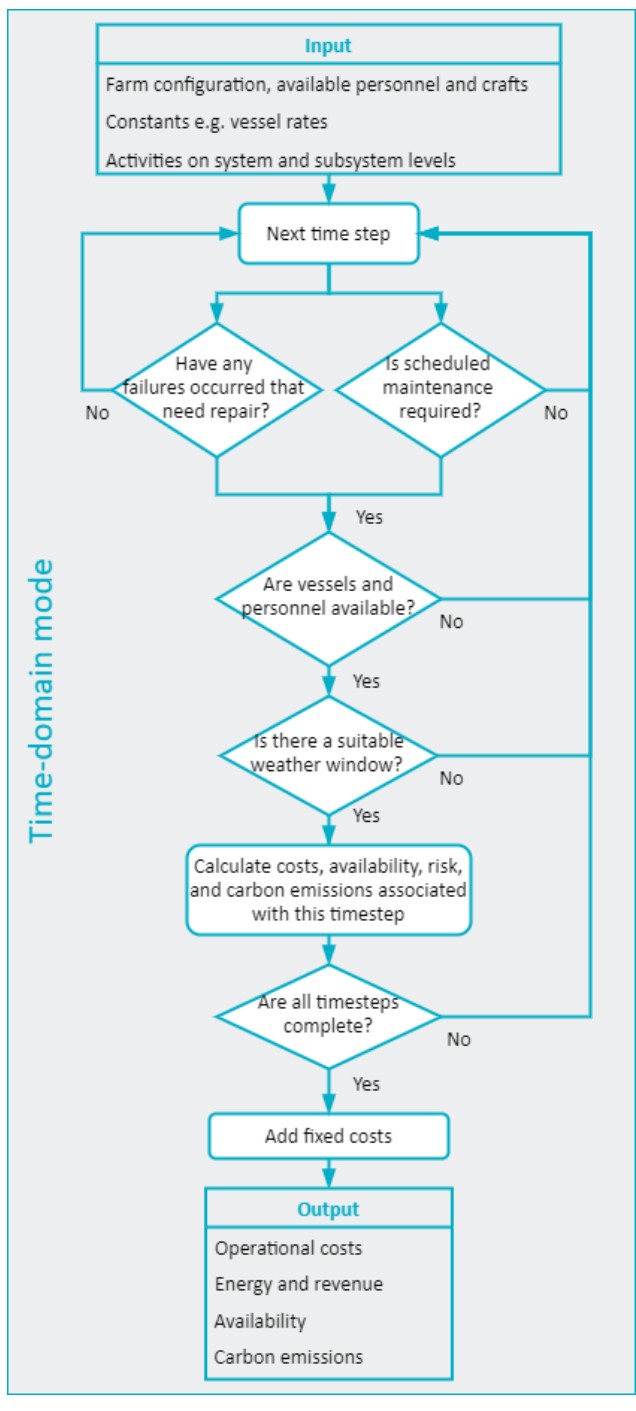

**Figure 1.** COMPASS tool workflow when operating in the time-domain mode.

outputs, the COMPASS tool calculates the overall risk rating associated with each activity and measures carbon emissions emitted by operation vessels which are summed up in the outputs of a simulation.

## 3.2  SOV and OMB modeling in COMPASS

Each activity in COMPASS has a list of required personnel to perform that activity, which was defined by a user in the inputs. This personnel can be of different type e.g. technician or a specialist. When an activity is due, a contractor gets created if it is not already available in the port. Each contractor gets assigned to a port, maintenance base or a SOV. In case if an activity requires a SOV then only the contractors from that SOV will be considered to perform this activity and if there is no personnel of the required type, a new contractor will be generated if it fits into the SOV accommodation. When SOV accommodation is

full and there is no personnel of the required type, the activity will not be completed. When a SOV returns to port, it empties its accommodation and the process of creating new contractors starts from scratch according to the activities that are due next. This way the code can guide a farm operator on the optimal type and the number of personnel to put on a SOV.

When activities accumulate they get scheduled by the software tool according to the urgency. If two assets have activities due with the same urgency then the asset with more activities due is prioritised. When several activities are due on the same

asset and have the same vessel requirement they will be merged together until the vessel capacity (in terms of personnel) is reached.

When the task is complete and a personnel becomes available it can either move to the next task or be picked up by a suitable vessel and return to the place where it was generated (SOV, port or OMB). Each vessel will pick up as many personnel as its capacity allows from the nearest assets and will update the carbon emissions accordingly.

## 3.3  Net Present Value

For the duration of the SOV contract period it is expected that there will be a fixed regular fee paid monthly or annually to the vessel owner. Contract duration varies from project to project but typically it lasts several years. While the cost for the SOV will be distributed over the years, the fixed cost associated with building an OMB will occur before the farm goes live. Because of this difference in timing of the cash flow, Net Present Value (NPV) needs to be taken into consideration using Equation 1

where $C_t$ is the net cash flow at time $t$, $r$ is the discount rate and $t$ is the time of the cash flow (measured in years in this study).

$$NPV = \frac{C_t}{(1+r)^t} \tag{1}$$

## 4  Inputs and assumptions

### 4.1  Farm Configuration

The hypothetical floating wind farm used for this study is located off the North East (NE) coast of Scotland and consists of

sixty-six 15 MW semi-submersible floating wind turbines located with an approximate distance of 2 km to each other (in

**Table 1.** Characteristics of the case study floating wind farm

| Site name | NE8 |
|---|---|
| Number of turbines | 66 |
| Turbine capacity | 15 MW |
| Main O&M port | Peterhead |
| Distance to the port | 100 km |
| Maximum water depth | 100 m |

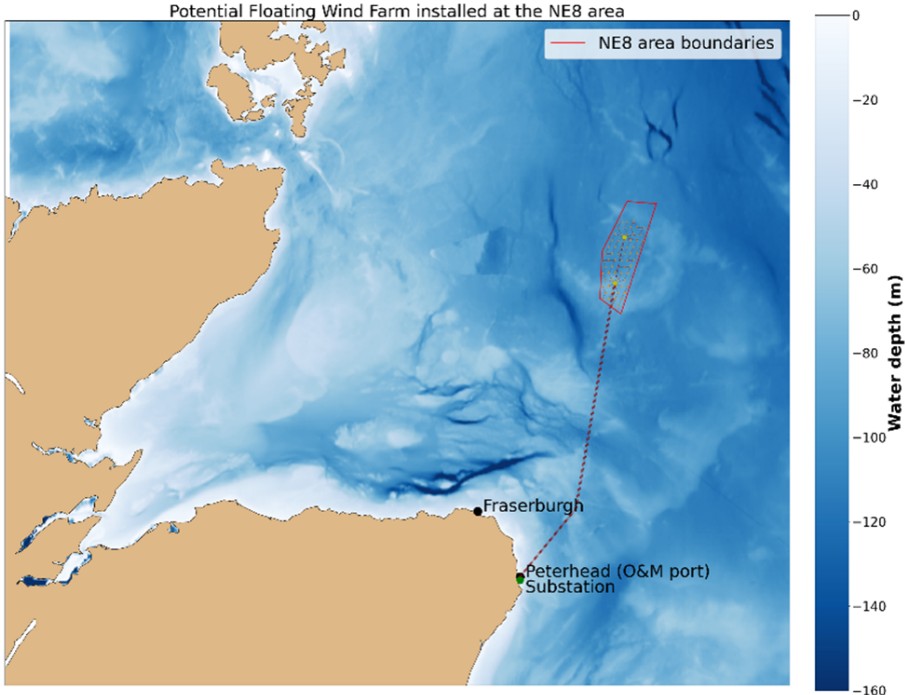

**Figure 2.** Hypothetical Floating Wind Farm Layout

latitudinal and longitudinal directions). The site is located within the boundaries of the NE8 seabed leasing area under the Scottish Government's 'sectoral marine plan for offshore wind energy' ('ScotWind'). This was highlighted as a site suitable for floating wind in the ORE Catapult Floating Wind Centre of Excellence's cost reduction report (ORE Catapult, 2021). Characteristics of the site are summarised in Table 1 and the approximate farm layout is shown in Fig. 2 and 3. This is not intended to replicate how the real wind farm layout in that area will look, as no environmental measurements nor wake assessments have been performed. However, it is sufficient for O&M simulation.

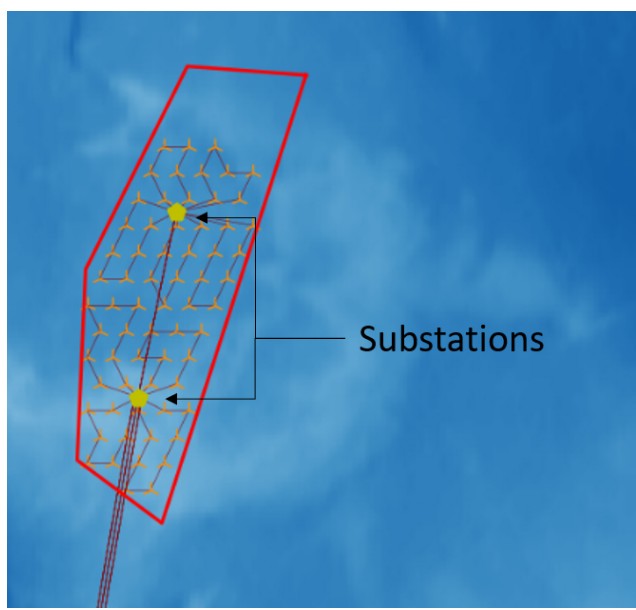

**Figure 3.** Close-up view of the farm

It was assumed the capacity of each offshore substation will be 500 MW. Therefore, two substations would be required for this site. These would be connected to the onshore substation in Peterhead by two export cables. Fraserburgh and Peterhead are the closest O&M ports to the considered wind farm, however, Peterhead has the capacity to play a role in assembly and
manufacturing and is therefore selected as the main O&M port for this farm. It was assumed that all repairs and maintenance in this farm are to be carried out offshore. The use of CTVs may be unfeasible for this site, given that the vessels would have to travel approximately two hours to get to the farm.

## 4.2 Weather Data

One of the inputs required to run COMPASS is weather data. There are two weather datasets that were used for this study:
ERA5 and ERA-20C, both are open-source global reanalysis data. ERA5 provides hourly data running until 2019 which can be retrieved from the Copernicus website (Copernicus, 2018), it gets updated yearly based on new observations. ERA-20C data was retrieved from (ECMWF, 2010), it is only available until 2010 and provides data for every 3 h interval. ERA5 data however has lower spatial resolution than the ERA-20C data, the respective resolution is 0.5 and 0.125 in both longitude and latitude. COMPASS uses 1 h intervals in its time-domain mode, therefore ERA-20C data was linearly interpolated to fill in 2 h gaps.
Figure 4 shows the comparison between the two datasets. The data in the Fig. 4 was averaged over 25 years and 1990-2015 year span was used with ERA5 data and 1985-2010 year span with ERA-20C. It can be seen that peaks and troughs of the two datasets align well even though two datasets were retrieved for different coordinates. ERA-20C tends to model lower wave height resulting in 1.82 m mean wave height compared to 1.95 m for the ERA5. ERA5 mean wave height matches better with

other sources for the same location and with how climate is changing it is important to have up-to-date data to model accurate results. Nevertheless, both data sets were used in this study to see how the choice of data affects the outputs.

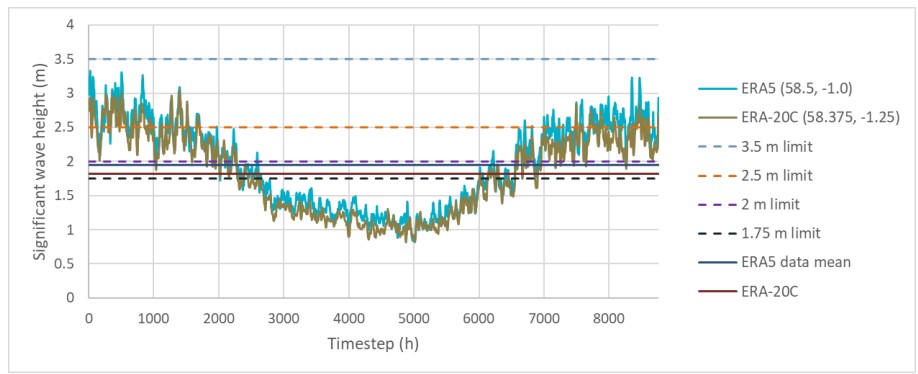

**Figure 4.** ERA5 and ERA-20C reanalysis data averaged over 25 years for each timestep


Figure 5 shows the number of opportunities in January to finish an activity. Currently tasks in COMPASS are not interruptable meaning that the software will look for a 12 h weather window if a 12-hour-long activity occurs, however making that change may increase the availability of the farm due to increased number of weather windows. The effect of energy availability on the OMB strategy results is demonstrated in Sect. 5.

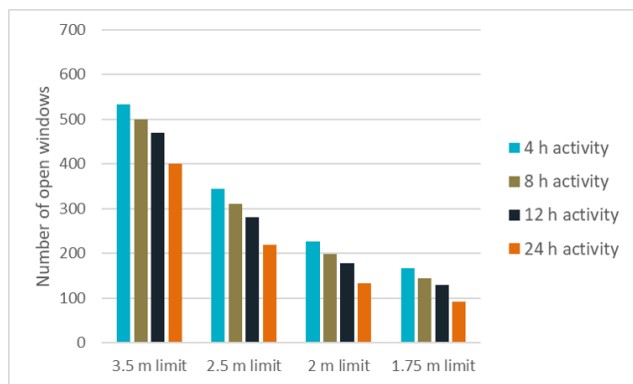

**Figure 5.** Number of weather windows for each activity duration in January for different wave limits (bottom axis)

**4.3 Floating Wind Unplanned Maintenance**

There is a lack of actual reliability data from floating wind turbines due, in part, to limited deployment to date. Therefore, most failure rates are inferred from the oil & gas industry, reliability data from ships, and other research areas. There have also been studies estimating the floating wind component failure rates using Bayesian networks (Li et al., 2020). The future generation floating turbines are likely to be hybrid moorings having a chain-synthetic mooring-chain configuration for several

reasons including reduced loads, better fatigue life and durability (Weller et al., 2015), reduced corrosion and easier installation and maintenance due to the lighter weight of these moorings. The synthetic mooring failure rate was estimated from the DTOcean+ project (DTOcean, 2015) and was combined from the "Polyester rope", two "Connectors" and "Other" components' failure rates, assuming that the chain part of the mooring does not contribute to its failure rate (as a large proportion of the chain will lie on the seabed where the tension is significantly lower than in the hanging section of the mooring) (Borg et al., 2020). This resulted in the failure rate presented in Table 2 which is lower than that for a chain mooring. This was expected as synthetic ropes are resistant to corrosion and have a much greater fatigue life (chain mooring failure rate is 0.0025-0.00378 (Fontaine et al., 2014; DTOcean, 2015)). The anchor failure rate was taken from a reliability study of drag embedment anchors (Javad Moharrami and Shiri, 2018) which are most commonly used for semi-submersible platforms. Structural damage frequency was assumed to be the same as it is in the oil industry data for mobile platforms (Moan, 2009). The array cable failure rate was updated according to the latest research and the distance between turbines (Warnock et al., 2019). Failure data for dynamic cables does not yet exist and, whilst cable manufacturers claim the fatigue life of dynamic cables is over 25 years, there are failures which may occur accidentally due to collision, harsh weather and damage during installation and manufacturing. It was assumed that the dynamic cable failure rate is twice as high as for the array cable because of harsher environment it is susceptible to.

In four out of the five components listed in Table 2, a complete replacement is assumed to be required after a failure occurs. The replacement costs are listed in Table 2 and are assumed to be the same as the original components costs. Full replacement is assumed for all dynamic cables and array cables in case of a failure. What makes it unfeasible to repair a section of a cable instead of a full replacement is the complex and time-consuming inspections of a failed cable to identify the failure point as well as long cure hours to connect the two ends of a cable and the risk of a further damage during the repair process. The most common cable connection is the dry connection which means a cable would be required to be lifted out of the sea for repair. The risks and the costs associated with the downtime required for array cable section repair outweigh the cost savings from repairing just this section. However in case of an export cable which is much longer it can be cheaper to repair a part of the export cable than to replace the entire cable. Costs of anchors may range up to £225,000 (James and Ros, 2015), however the drag-embedment anchors are usually the cheapest. The price for the drag-embedment anchor is assumed to be £60,000 according to the Carbon Trust report (James and Ros, 2015), which is similar to what is given in the DTOcean report (DTOcean, 2015) of £67,900. Prices for full chain moorings according to the DTOcean vary between 126 and 233 £/m, and are 600 £/m according to the Carbon Trust report. According to the Carbon Trust the cost of the hybrid mooring would be 2200 £/m however this cost is likely to vary depending on the type of plastics used in the mooring and its width which is likely to vary depending on the dynamics of the sea and water depth. Because a semi-submersible floating foundation is assumed, the length of the mooring is considered to be 6 times the water depth. The cost of replacing a dynamic cable was calculated using the coefficients provided in the Corewind report (Ikhennicheu et al., 2020) for a 33 kV dynamic cable resulting in 385 £/m assuming the length of the dynamic cable is $(2 \times$ water depth (m)$\times 2.6)$. 33 kV is the maximum cable rating for which the cost coefficients were provided and is currently a common rating for offshore wind farm cables, however in the future we might see more 66 kV cables which are more expensive but lead to the reduction of the cost of energy (Ferguson et al., 2012).

**Table 2.** Failure rates and replacement costs of floating wind turbine subsea components assumed for this study

| Component | Failure Rate (failures/component/year) | Cost (£) (per component) |
|---|---|---|
| Hybrid synthetic mooring | 0.0017 (per km) | 520,000 |
| Anchor | 0.00012 | 67,900 |
| Semi-submersible platform (structure damage) | 0.018 | - |
| Array Cable | 0.003 (per km) | - |
| Dynamic Cable | 0.003 | 200,000 |

## 4.4 Floating Wind Planned Maintenance

The annual survey is assumed to be carried out with an observation ROV to visually check the condition of all moorings. For a special survey (i.e. once every five years) it is recommended where possible to raise moorings to the surface for a more detailed inspection (Ma et al., 2019). It is assumed that half of the special surveys will be carried out with a vessel onto which a mooring will be raised, and the other half with an observation ROV (i.e. subsea). Periodic inspections of moorings may involve checking an angle of the catenary mooring to estimate the changes in tension, which can also be done by an observation ROV. Anchors are expected to be inspected together with the moorings, anchor failure rates are very low (Table 2) and so do not require any additional planned inspection.

Dynamic cables require regular (every couple of years) visual inspection of bending stiffeners and hang-offs, any hardware attached to the cable free span, buoyancy modules and cables themselves (for marine growth), protection sleeves and transition joints (Jensen et al., 2015). All dynamic cable inspections are assumed to be done with an observation ROV to reduce the deployment of divers and decrease health and safety risks, because water depth in the site area reaches 100 m in some locations. Marine growth removal every 5 years was added as a regular activity to remove any additional weight which may increase the tension in hanging cables, especially in the top 10-15 m of the cable.

## 4.5 Fleet Assumptions

For the first strategy one SOV was used with one daughter craft and a capacity to accommodate 40 technicians. It was tested with additional simulations that increasing the number of SOVs does not have a significant effect on the availability but can increase costs significantly. Depending on the dynamic positioning technology and the gangway technology the limits for technician transfer can vary between 2.5 m and 4.5 m however 3-3.5 m are most common limits (Hu and Yung, 2020). Some SOVs may have more daughter crafts, however for this study a medium size SOV was assumed based on the North Star SOV. For the second strategy a fleet with three CTVs was assumed, increasing the fleet size did not have any significant effect on the availability of the farm. CTVs can be different in size and design and vary in their way of attachement to the turbine. This makes it hard to come up with one representative value for the capacity and wave limits. Although next generation CTVs

**Table 3.** Characteristics of the OMB scenarios simulated in this study

| Scenario name | OMB 1.75 m ERA5 | OMB 2.5 m ERA5 | OMB 1.75 m ERA-20C | OMB 2.5 m ERA-20C |
| --- | --- | --- | --- | --- |
| Weather data source | ERA5 | ERA5 | ERA-20C | ERA-20C |
| CTV transit limit | 1.75 m | 2.5 m | 1.75 m | 2.5 m |
| CTV transfer limit | 1.75 m | 1.75 m | 1.75 m | 1.75 m |

**Table 4.** Characteristics of the SOV scenarios simulated in this study

| Scenario name | SOV 3.5 m ERA5 | SOV 3.5 m ERA-20C |
| --- | --- | --- |
| Weather data source | ERA5 | ERA-20C |
| SOV transfer limit | 3.5 m | 3.5 m |
| Daughter Craft transfer limit | 1.75 m | 1.75 m |

may be bigger in size and be able to sit 24 personnel, most common CTVs have a capacity to transit 12 technicians. CTVs which could be installed on the OMB are expected to be smaller and hence the capacity of 12 was selected. Current generation CTVs can operate in 1.2-2.5 m waves (Stumpf and Hu, 2018), however according to experience due to sea sickness and safety concerns as well as steeper waves far offshore, most CTVs travel in 1.5-1.75 m waves. Future generation CTVs promise to have better motion compensating systems and designs to comfortably transfer personnel in 2.5 m waves. Due to this variability of wave limits two options were selected for this study with current generation CTVs and future generation CTVs. In total six scenarios were simulated in this study that are summarised in Tables 3 and 4.

## 5 Results

### 5.1 Convergence study

The fact that the Monte Carlo method is used in the software for modelling random failures means that each simulation will produce a different set of random numbers resulting in varying outputs from simulation to simulation. In order to get a good estimate of the range within which OPEX and other KPIs will lie, simulations need to run multiple times. The higher the number of simulations, the closer the output gets to the true mean of all KPIs. The level of confidence in results is determined by a convergence study. To perform the convergence study $95\%$ confidence level was selected. $95\%$ confidence interval bounds

can be calculated using Student's t-distribution, a type of distribution that is similar to normal distribution but is commonly used when a number of samples (simulations in this case) is low ($< 30$). Confidence interval was calculated using Equation 2

$$CI_{95\%} = \mu \pm SE \tag{2}$$

290  Where SE is the standard error calculated using Equation 3.

$$SE = t_{95\%,n-1}\frac{\sigma_n}{\sqrt{n}} \tag{3}$$

In Equation 3, $\mu$ is the mean value of the sample, $t$ is the confidence level value which varies with the confidence level and number of samples (simulations in this case), $\sigma$ is the standard deviation of the sample population and $n$ is the sample size. The value of $t$ can be calculated numerically, however in this study CONFIDENCE.T() function was used in Excel to calculate 295  the $95\%$ confidence interval using built-in $t$ values.

The convergence study presented in Fig. 6 shows how percentage error of OPEX results mean converges with increasing the number of simulations. The error in the graph represents the proportion of SE of the mean of simulations. Larger number of simulations would lead to more accurate results however they would require more computational time especially because convergence slows down with the number of simulations. The standard error of $2\%$ of the mean was considered acceptable in 300  this study. It can be seen that running 20 simulations is enough to bring the error below $2\%$.

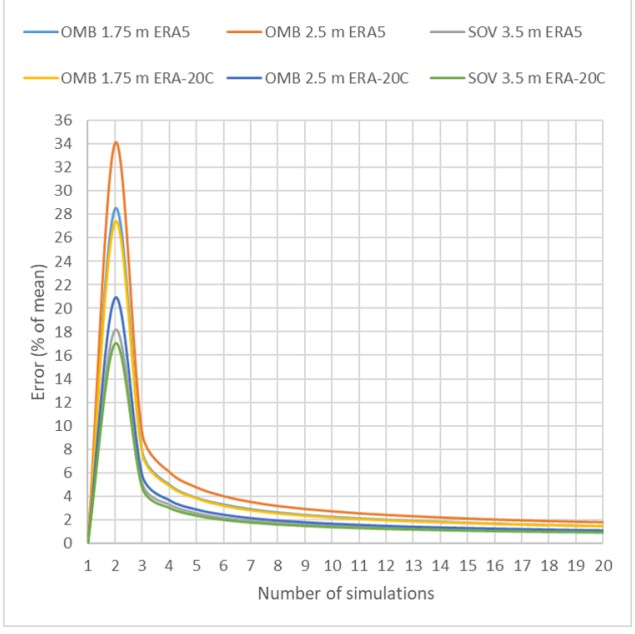

**Figure 6.** OPEX convergence after running 20 simulations of each scenario

**Table 5.** OPEX, time availability (TA) and energy availability (EA) results from all six simulated scenarios.

|  | OPEX (£/kW) | OPEX Error (%) | TA (%) | TA Error(%) | EA (%) | EA Error (%) |
|---|---|---|---|---|---|---|
| OMB 1.75 m ERA5 | 46.21 | 1.43 | 92.06 | 0.13 | 89.64 | 0.12 |
| OMB 2.5 m ERA5 | 46.67 | 1.76 | 94.22 | 0.08 | 92.77 | 0.09 |
| SOV 3.5 m ERA5 | 49.01 | 0.96 | 97.12 | 0.02 | 97.18 | 0.04 |
| OMB 1.75 m ERA-20C | 45.13 | 1.42 | 92.47 | 0.17 | 89.73 | 0.19 |
| OMB 2.5 m ERA-20C | 44.63 | 1.07 | 94.43 | 0.17 | 92.74 | 0.13 |
| SOV 3.5 m ERA-20C | 48.57 | 0.89 | 97.33 | 0.02 | 96.93 | 0.03 |

## 5.2 Simulation Results

Table 5 summarises the 6 types of simulations each of which were run 20 times to minimize the error in the mean outputs. The table shows the results for the total OPEX of these simulations which includes fixed costs, the costs of hiring the vessels and personnel costs as well as the average time and energy availability resulting from each scenario over all years. The cost associated with building the OMB are not included in this table in order to compare OPEX only. The percentage error presented in the table was based on the convergence study i.e. it is an error calculated using Equation 3 based on the results of 20 simulations. Because fixed long-term contracts are assumed for SOVs and CTVs the uncertainty in results arises mostly from the frequency and costs of maintenance activities generated with a Monte Carlo method. Therefore, when considering the difference between two strategies, the error effect on that difference will be small because e.g. if an upper bound is taken for the OMB scenarios, the upper bound should also be taken for the SOV scenarios. Previous studies (listed in Table 6) have been used to benchmark the results of this study. Results lie between the NREL minimum and maximum estimations (Musial et al., 2020) but are much lower than other estimations. Compared to the estimations presented in Table 6 COMPASS outputs are produced using much more detailed inputs and simulations. Cases with a SOV resulted in the highest availability due to higher weather limit but drawback of this strategy is the higher OPEX due to significantly higher costs of hiring these vessels compared to CTVs. In can be seen in Table 5 and later in Fig. 7 that the 0.13 m difference in wave means of two weather datasets ERA5 and ERA-20C affects the difference between results making the cases with ERA5 data more expensive. This difference is to do with the fact that vessels (e.g. ROV support vessel or jack-up) return to port more often with ERA5 data than with ERA-20C due to slightly worse sea conditions, however further improvements to the software should reduce that difference. There is also an effect of having lower time resolution in ERA-20C data which is linearly interpolated to model hourly data. ERA5 is hence more restrictive to the vessel logistics because it allows for fluctuations in wave height within each 3 h time spans. Energy availability however varies less than time availability between scenarios with the same logistical setup and different weather datasets. This is to do with the fact that ERA-20C not only predicts lower waves in that location but also lower winds. It should be noticed however that time availability results are more sensitive to switching from one dataset to another for cases with lower wave limits for vessels.

**Table 6.** O&M cost estimations from various reports for comparison with the results from simulations.

|  | OPEX (£/kW) |
| --- | --- |
| (Myhr et al., 2014) | 114 |
| (James and Ros, 2015) (Commercial) | 89 |
| (James and Ros, 2015) (Pre-commercial) | 139 |
| (Musial et al., 2020) (min) | 27 |
| (Musial et al., 2020) (max) | 59 |

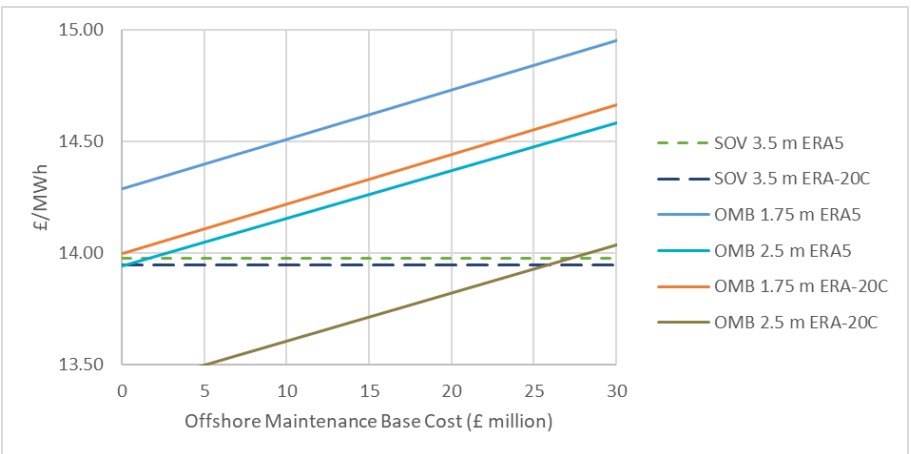

**Figure 7.** Six scenarios compared with varying the costs associated with the OMB

NPV formula shown in Equation 1 was applied on both the regular SOV hiring fee and the energy produced throughout 25 years. It was assumed that after SOV contract expiration it is renewed with the same pricing, however in future work the rise/drop of the fee should be taken into account. For the OMB scenarios the cost of building the base was varied between 0 and 30 £ million in order to find the break-even points at which having the OMB would be cost effective. The discount rate was assumed to be $5.5\%$ estimated by ORE Catapult (2021) for the first commercial-scale projects in UK. Figure 7 shows the results of this analysis. For the cases with the OMB and low wave limit, even when the capital cost associated with the OMB is low the lines stay above the SOV scenarios. The big difference between ERA5 and ERA-20C cases for OMB scenarios is to do with the fact that due to worse weather conditions vessels return to the port more often and hence their usage is longer in scenarios with ERA5 data. With ERA-20C data vessels can more often maintain several assets in a row without returning to port. Scenarios with high wave limit i.e. 2.0 m for transfer and 2.5 m for transit are the only ones that have a section of the line below SOV scenarios.

As was mentioned in Sect. 4.2 future work in COMPASS includes making activities suspendable. Splitting up activities can potentially increase time and energy availability. If a wind farm was installed at a different location with calmer weather

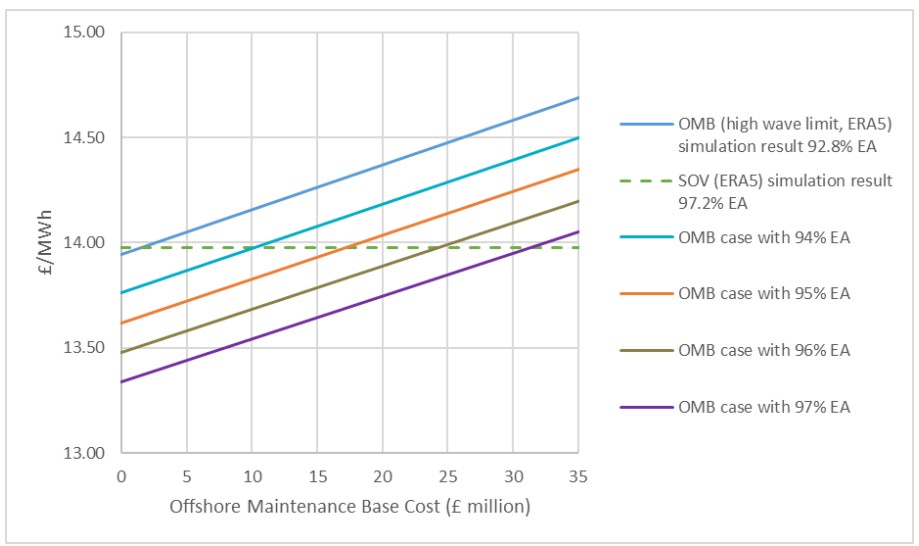

**Figure 8.** The effect of changing energy availability on the results

conditions the availability could also be higher. Figure 8 assumes that OPEX costs stay the same but energy availability increases for the scenario with the OMB, high wave limit and ERA5 data. In the bottom four lines more energy is produced
340 at the same cost which results in lower price per MWh. Even for the case with the highest availability presented in the graph the OMB would need to contribute less than 32 £ million to the total CAPEX which is unlikely if the OMB has a separate substructure. According to the report published by the ORE Catapult (Jump et al., 2021) a separate substructure of this size would cost at least 49.5 £ million. Appendix A shows how the results in Figures 7 and 8 would change if the discount rate was 4.2% that is estimated for floating offshore projects when the industry matures. Changing the discount rate only affects
345 the OMB results. Decreasing the discount does not change the origins of the lines representing OMB scenarios but makes their slope more gradual thus moving the break-even point to the right on each figure.

Spare part storage capacity is not considered in both scenarios, however in the SOV case modeling the additional return to port would not affect the costs but would change the carbon emissions and availability output. This is due to the assumption of a long-term SOV contract. However in the OMB case there would be an additional cost of helicopter or vessel delivery of a
350 part if it is not available at the site.

Carbon emissions were measured in each simulation and averaged out for SOV and OMB scenarios. Table 7 provides the results from this study and benchmarks them over the most recent studies. The results align well with the Sustainability report results published by Orsted (2020) however they are expected to be higher as further changes to COMPASS are incorporated. The emissions resulting from stand-by of the vessels or dynamic positioning are not included here, however emissions due to
355 transit are expected to be the highest in these three categories.

**Table 7.** Benchmarking results from simulations with other studies (Gray, 2021b)

| Case Study | Carbon Emissions (g/kWh) |
|---|---|
| SOV cases | 1.96 |
| OMB cases | 1.3 |
| ORE Catapult (SOV only) | 2.1 |
| Arvesen et al. (2013) | 4.93 |
| Orsted (2019) | 3.5 |
| Orsted (2020) | 1.6 |

## 6 Conclusions

This study analysed two logistical strategies that could potentially be used for offshore floating wind farms located over 70 km from the main onshore O&M port. Two strategies were modelled using ORE Catapult's in-house O&M simulation tool, COMPASS; one with a SOV performing most of the maintenance and the other utilising an Offshore Maintenance Base (OMB) accommodating three CTVs. Sensitivity studies were undertaken on varying wave limits of the CTVs and two weather data sets ERA5 and ERA-20C. A case study with sixty-six 15 MW turbines on semi-submersible platforms was used. The results of this work showed that having one SOV is a preferable strategy if costs and energy are taken into consideration unless the OMB shares its foundation with the substation. Further work is required to analyse the increase in foundation cost due to additional structures. OPEX and carbon emissions calculated using simulations were compared with other studies and correlate well with some of them. Carbon emissions are 34% lower in scenarios with OMB than with a SOV and that difference is expected to be even higher if fuel usage during stand-by and dynamic positioning is included in COMPASS. It was also found that the results are sensitive to the weather data used: where ERA5 data is used, vessels tend to return to port more often thus increasing the costs. The difference between scenarios is however expected to reduce if further work is implemented in COMPASS. This includes making tasks suspendable, building in tow-to-port capability in COMPASS, and taking into account work shifts of personnel. The range of OPEX costs collected from different studies is too wide to evaluate the accuracy of the costs produced by COMPASS with confidence, therefore further work includes verification of the tool against other tools and validation against real life O&M where possible. Future work also includes further calibration of failure rates, repair time and costs according to the latest available data from floating wind industry and incorporating a failure rate distribution to model a higher chance of failure in the beginning of a project due to damage during installation.

*Author contributions.* NA conceptualised this research supervised by AG, IL, GR, RCT. NA developed the methodology with the support from AG, particularly in calculating the Net Present Value. AG and NA programmed the time-domain mode in the COMPASS tool and

NA collected the required input data and performed the simulations. NA performed the formal analysis and AG provided the benchmarking material. IL, GR, RCT supervised this research activity and reviewed the published work.

*Competing interests.*   No competing interests are present.

*Acknowledgements.*   This research is funded by the EPSRC and NERC for the Industrial CDT in Offshore Renewable Energy (EP/S023933/1) and sponsored by the ORE Catapult.

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

**Appendix A**

Figures A1 and A2 show what the results would be if the discount rate was $4.2\%$

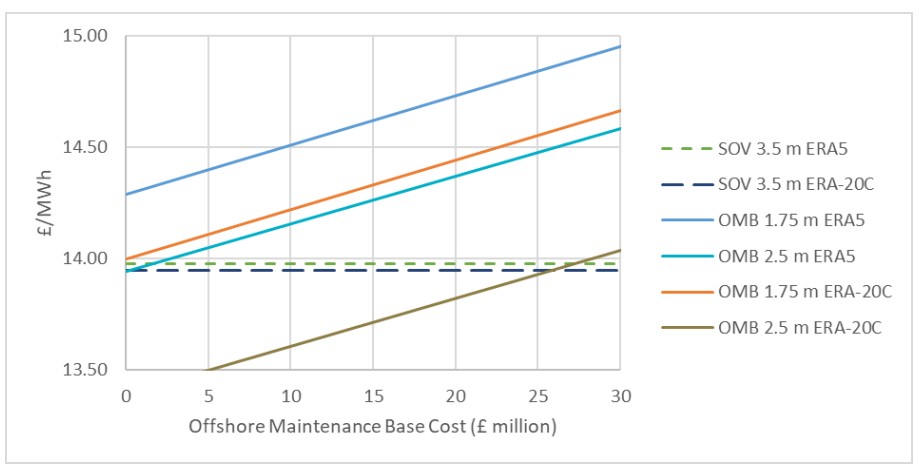

**Figure A1.** Six scenarios compared with varying the costs associated with the OMB

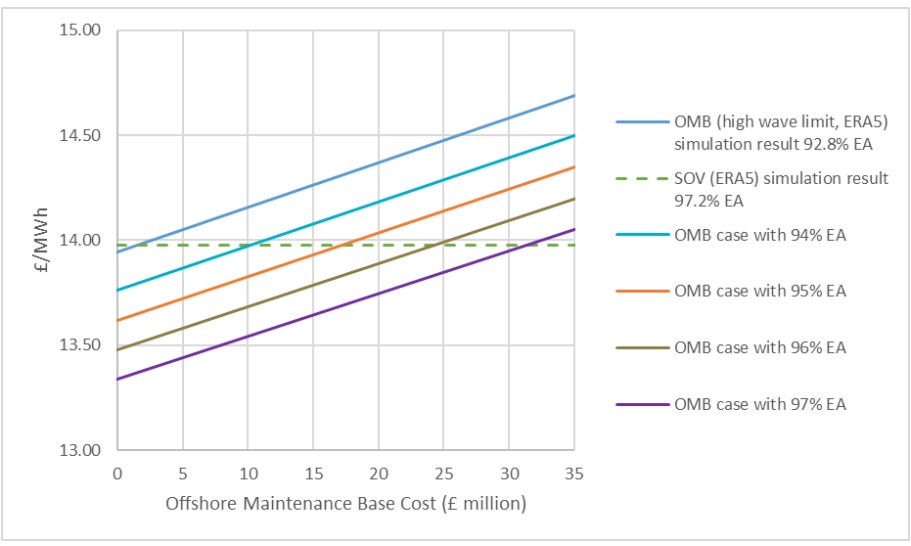

**Figure A2.** The effect of changing energy availability on the results