# Peer review of "Analysing the effectiveness of different offshore maintenance base options for floating wind farms"

_Wind Energy Science, 2021_

## Author Response (AR1)

**Response to RC1**

Thank you for taking your time and reviewing our paper. The feedback you provided is very clear and helpful. Here are the responds to your comments:

Specific comment 1:

This is a very good comment as the calculation of the 95% confidence intervals is not so straightforward. In fact, after the review of the comment the calculation was checked and the decision was made to adjust the formula for calculating the CI because a number of samples (i.e. simulations) is small (20) a Student's t-distribution should be used rather than a normal distribution which leads to a different way of calculating the CI. In the finalised version of the paper the formula to calculate the relative error is explicitly presented and explained (Sect. 5.1).

Specific comment 2:

   a) The percentage error is that of the OPEX. It is now clarified in the finalised paper (l. 304).
   b) "O&M costs" has been changed to OPEX throughout.
   c) The availability error is included now in the finalised version (Table 5). It was not included before because it converges much faster than the OPEX. This is why it was OPEX error that was used to analyse convergence.
   d) A brief comment about the error effect on the differences between simulated cases is included now in the finalised version (l. 307).

Comment 3:

Mentioned vessels have different characteristics from SOVs and have therefore not been considered or mentioned in the paper. These vessels either miss a gangway or have a smaller accommodation capacity or lower wave height limit. The recommended paper [D] is a very interesting study and is now mentioned in the finalised version of the paper (l. 110).

Comment 4:

Although the recommended reference is very insightful, it is not included in the finalised version of the paper because it did not consider SOVs or offshore maintenance bases. The information on varying the minimum working time is also quite limited in that reference with only slight differences shown for the 1 hour and 2 hour minimum working time. Because our study did not focus much on varying the working time, it was decided to not include the recommended reference in it.

Comment 5:

This is a very good comment and a valid suggestion. It is indeed true that there is more fluctuation in ERA5 data which affects the difference between the results. The suggestion is now included in the finalised version of the paper (l. 318).

Technical corrections:

All technical corrections have been included in the finalised version of the paper.

**Response to RC2:**

Thank you for taking your time and reviewing our paper. The feedback you provided is very useful and the paper was edited where possible according to the recommendations provided. Here is the response to your comments:

- Electric system is included in the inputs for all simulations. Each turbine in COMPASS is split into subsystems e.g. Generator System or Control and Protection System. However activities for these systems are not discussed in the paper mainly because they have been created by the ORE Catapult team and are not publically available. The final version of the paper includes more detail on how turbines are split into subsystems and components (l. 133).
- The sensitivity of the results against maintenance costs is not included. If maintenance costs would change equally for both strategies there would be no effect on the difference between them. SOV cost variation is not considered significant (especially in comparison with other factors) and is therefore is not included in this study.
- The discount rate has been changed according to the latest findings (l. 327). An Appendix A was added to the final version of the paper which shows how the results would change if a discount rate was different.
- Spare part storage has not been considered in this paper and it is now explicitly mentioned in the finalised version of the paper. A reference for interested readers has also been added which analyses storage capabilities of SOVs (l.110).
- The use of commas was revised.